# Benefits of Polyamide Nanofibrous Materials: Antibacterial Activity and Retention Ability for *Staphylococcus Aureus*

**DOI:** 10.3390/nano11020480

**Published:** 2021-02-13

**Authors:** Simona Lencova, Kamila Zdenkova, Vera Jencova, Katerina Demnerova, Klara Zemanova, Radka Kolackova, Kristyna Hozdova, Hana Stiborova

**Affiliations:** 1Department of Biochemistry and Microbiology, Faculty of Food and Biochemical Technology, University of Chemistry and Technology, Technicka 3, 16628 Prague 6, Czech Republic; zdenkovk@vscht.cz (K.Z.); demnerok@vscht.cz (K.D.); zemanovl@vscht.cz (K.Z.); kolackor@vscht.cz (R.K.); 2Faculty of Science, Humanities and Education, Technical University of Liberec, Studentska 1402/2, 461 17 Liberec 1, Czech Republic; vera.jencova@tul.cz; 3Elmarco Ltd., Svarovska 621, 460 01 Liberec, Czech Republic; kristyna.hozdova@elmarco.com

**Keywords:** polyamide, nanofiber, permeability, filtration apparatus, antimicrobial activity, *Staphylococcus aureus*, long-term stability

## Abstract

Although nanomaterials are used in many fields, little is known about the fundamental interactions between nanomaterials and microorganisms. To test antimicrobial properties and retention ability, 13 electrospun polyamide (PA) nanomaterials with different morphology and functionalization with various concentrations of AgNO_3_ and chlorhexidine (CHX) were analyzed. *Staphylococcus aureus* CCM 4516 was used to verify the designed nanomaterials’ inhibition and permeability assays. All functionalized PAs suppressed bacterial growth, and the most effective antimicrobial nanomaterial was evaluated to be PA 12% with 4.0 wt% CHX (inhibition zones: 2.9 ± 0.2 mm; log_10_ suppression: 8.9 ± 0.0; inhibitory rate: 100.0%). Furthermore, the long-term stability of all functionalized PAs was tested. These nanomaterials can be stored at least nine months after their preparation without losing their antibacterial effect. A filtration apparatus was constructed for testing the retention of PAs. All of the PAs effectively retained the filtered bacteria with log_10_ removal of 3.3–6.8 and a retention rate of 96.7–100.0%. Surface density significantly influenced the retention efficiency of PAs (*p* ≤ 0.01), while the effect of fiber diameter was not confirmed (*p* ≥ 0.05). Due to their stability, retention, and antimicrobial properties, they can serve as a model for medical or filtration applications.

## 1. Introduction

Skin, being the largest organ of the human body, is exposed to various influences. From ancient times, skin injuries are among the most common types of health issues, resulting in the need for reliable treatment. The evolution of wound treatment led from materials that only stopped the bleeding to the development of functionalized materials with an antibacterial effect to prevent infections [1].

A wide range of pathogenic microorganisms can cause infections [2]. Furthermore, the microbiota present in the wound change during the infection process. In its initial stage, Gram-positive bacteria *Staphylococcus aureus* and *Streptococcus pyogenes* predominate, while Gram-negative bacteria, such as *Escherichia coli* and *Pseudomonas aeruginosa*, are usually detected in the final stages of infections and chronic wounds [3,4]. All of these bacteria can cause severe complications, and, therefore, various materials are used as wound dressings to reduce the risks associated with such infections.

These materials are expected to have specific characteristics. The ideal wound dressing should not only be a reliable barrier against foreign elements from the external environment but should also keep the injury moist, allow the exchange of gases, act against microorganisms, and remove redundant exudate. Thus, materials used as wound dressing must be safe for the patient, which means they must be non-toxic, non-allergenic, and either biodegradable or easily removable from the wound [1,5,6]. Fulfilling all of these requirements is not an easy task; therefore, recent research has focused on the development of functional wound dressings.

Nanomaterials with their unique properties can be very beneficial in this field. They make it possible to control external conditions at the injury site to ensure optimal cellular activity during the healing process, mostly adequate air access while preventing unwanted elements and microorganisms from the environment from entering the wound [6,7]. By fulfilling all of these requirements, nanomaterials are expected to have a barrier capability. Because of this, they can be used in the treatment of not only acute but also chronic skin injuries, in which the surface infections may result in a subcutaneous tissue infection and a directly life-threatening condition [1].

For a material qualify as a nanomaterial, it is necessary to know basic structural parameters. Nanomaterials differ in their behavior, depending on their fundamental characteristics resulting from the preparation process, even if they are made from the same polymer. Fiber diameter, surface density, air permeability, and fiber density (nanomaterial porosity) are the main characteristics [8]. Because of these parameters, materials differ in morphology, which is expected to affect their final properties, including the ability to act as a barrier. Nanofibrous materials were proven to be an effective barrier in several studies dealing with microbial filtration [9,10,11,12,13].

In addition to injury barriers, nanomaterials can act as carriers of substances with proven antimicrobial activity and thus inhibit the growth of microorganisms that are present. Nowadays, research is primarily focused on dressing materials for skin injuries enriched with various antimicrobial agents. The most common are as follows:Antibiotics, such as ciprofloxacin [14], gentamicin [15], or tetracycline [16];Nanoparticles (NPs), such as Ag [17,18], ZnO [19], or TiO_2_ [20];Natural extracts, such as curcumin [21] or *Hypericum perforatum* [22];Antiseptics, such as chlorhexidine (CHX) [23,24].

Silver and CHX are good examples of antimicrobial substances suitable for this purpose. Silver, with its proven antimicrobial properties against both Gram-positive and Gram-negative bacteria [25,26], has enormous potential for treating skin wounds infected by resistant bacteria, for example, the methicillin-resistant strain of *S. aureus* [17]. CHX acts bactericidally by destroying bacteria cell membranes [27], and due to its antiseptic properties, it is mainly used in medicine and oral hygiene as a component in mouthwash [28]. Its antimicrobial effect in wound dressings, which was more significant for Gram-positive bacteria, has also been confirmed by several studies [23,29,30,31].

For an optimal antimicrobial effect of functionalized nanomaterials, it is crucial to have a homogenous nanofibrous structure and, at the same time, homogenous distribution of the antimicrobial compound. Polymers used for nanomaterial production vary in their ability to form homogenous materials. In general, functional wound dressings are usually made of biodegradable polyesters, such as polycaprolactone (PCL) or polylactic acid, with the main advantage of being used on the wound until their complete absorption [32]. However, it is quite challenging to obtain a homogenous structure in the production of polyester materials. On the other hand, uniform nanofibrous materials from polyamide (PA), a non-degradable synthetic polymer, can be prepared more easily and are considered to be more homogenous than, for example, PCL [33,34,35]. Due to their high mechanical resistance, PA materials are used in food packaging, the manufacture of joint replacements, and wound treatments in medicine [6,32,36].

Because PA nanomaterials have considerable potential for use in the treatment of skin injuries, it is necessary to study microbial interactions with them before their common usage in healthcare. In our previous study [37], we reported how biofilm formation is influenced by the PA nanomaterials’ morphology and AgNO_3_ functionalization, because biofilm development poses a major threat to wound healing. In this follow-up study, we aimed to analyze the other properties of PAs that reduce the risks of developing an infection. We designed a methodology for analyzing the materials’ permeability for bacteria. Additionally, we tested the antibacterial activity and long-term stability of both nonfunctionalized and functionalized PAs with AgNO_3_ and CHX, which has been tested sporadically in connection with PA nanofibers [38,39]. CHX-functionalized PAs were prepared and characterized within this study, the other PAs were prepared and characterized as part of our previous study [37]. The bacterium *S. aureus*, being a common part of natural skin microbiota and a frequent pathogen in skin wound infections, was chosen as a model microorganism. To our knowledge, we are the first who publish basic research about the retention ability of PA nanomaterials for a clinically relevant bacterium and, thus, increase their usage in several medical applications. Furthermore, the presented comprehensive design of testing microbiological safety and stability of PA nanofibrous materials is unique.

## 2. Materials and Methods

### 2.1. Nanomaterial Preparation

Nanofibrous PA materials were prepared from polymer solutions using a needleless electrospinning method in a Nanospider™ NS 1 S500 U device (Elmarco, Liberec, Czech Republic) as described in the study by Lencova et al. [37]. The electrospinning of the CHX-added layers was conducted from solutions containing 2.0 and 4.0% (*w*/*w*) chlorhexidine dihydrochloride (TCI chemicals, Tokyo, Japan). CHX was added to the prepared 12% PA solution and stirred at room temperature until its complete dissolution.

### 2.2. Nanomaterial Characterization

All non-functionalized PAs and PAs functionalized with AgNO_3_ were characterized in terms of surface density, thickness, air permeability, and fiber diameter in the study of Lencova et al. [37]. The same parameters for CHX PAs were characterized in this study. Briefly, the surface density was evaluated from 10 × 10 cm samples taken from at least five different parts of the material; the thickness was measured with a Corp ID-C112XB device (Mitutoyo, Teplice, Czech Republic); air permeability was measured using a TEXTEST FX 3300 device (TexTest Instruments, Schwerzenbach, Switzerland). To evaluate fiber diameters, CHX PAs were sputter-coated with gold (14 nm), and pictures were taken using a scanning electron microscope (SEM) Tescan Vega3 SB Easy Probe (TESCAN, Brno, Czech Republic), and Nova NanoSEM 230 (Thermo Fisher Scientific, Waltham, MA, USA), respectively. The fiber diameter was evaluated from five pictures of each sample, at least 100 measurements from one sample in total, using the software NIS Elements (Nikon, Tokyo, Japan). Pore size of all 13 tested PAs was measured in this study; pore size was measured in a CFP-1200 AEL Capillary Flow Porometer (Porous Materials Inc., Ithaca, NY, USA) using the bubble point method.

### 2.3. Bacterial Suspensions

The Gram-positive bacterium *Staphylococcus aureus* CCM 4516 (eq. ATCC 6538), obtained from the Czech Collection of Microorganisms (CCM, Brno, Czech Republic), was used as a model microorganism in this study. *S. aureus* CCM 4516 is an isolate from a human lesion and is used as a control strain for testing disinfectants and antiseptics. Pure bacterial cells were resuspended in tryptone soy broth (TSB, Oxoid, Cheshire, UK) and then used for the experiments.

### 2.4. Inhbition Assay

A pure culture of *S. aureus* CCM 4516 was cultivated for 24 h at 37 °C in TSB. The suspension’s optical density (OD) was adjusted to 1 McFarland (McF). The tests were performed via the approaches described below.

#### 2.4.1. Inhibition Zone Method

For the inhibition zone method, we used a standard method adapted to the experiments. The plate count agar (PCA) plates were first inoculated by spreading five horizontal stripes of a bacterial suspension at a sufficient distance (approx. 1 cm) from each other, and a piece of a sterile PA (5 × 5 cm) was then placed onto the inoculated surface. At least three independent replicates were performed for each sample. After the incubation (24 h, 37 °C), inhibition zones (mm) that formed around the PA were measured (from the edge of the PA).

#### 2.4.2. Inhibitory Rate Method

A piece of a sterile PA (5 × 5 cm) was placed into a test tube containing 10 mL of bacterial suspension with an approx. concentration of 10^2^ CFU/mL prepared by the serial decimal dilution of the suspension with OD 1 McF. After incubation for 24 h at 37 °C, the changes were compared by quantifying CFU/mL; the suspension was serially decimally diluted. The obtained dilutions were applied in 20 μL droplets to a plate count agar (PCA, Oxoid, Cheshire, UK) in three parallels and incubated for 24 h at 37 °C. After the cultivation, grown bacterial colonies were counted and quantified [37]. Three independent replicates were performed for each sample. Bacterial suspensions without any added material were used as controls. Then, the inhibitory effect was calculated using the formula below (Equation (1)) [40]. From the CFU/mL determination, log_10_ suppression (CFU/mL) was assessed according to Equation (2) (log_10_ suppression expresses the difference between bacterial growth in the control and the suspension with the PA, both after incubation under defined conditions).
(1)$$\mathrm{Inhibitory}\;\mathrm{rate}\;\left[\%\right]=\frac{\mathrm{CFU}\left(\mathrm{control}\right)-\mathrm{CFU}\left(\mathrm{sample}\right)}{\mathrm{CFU}\left(\mathrm{control}\right)}\times 100$$
where CFU (control) is the number of CFU/mL in the bacterial suspension itself and CFU(sample) is the number of CFU/mL in the bacterial suspension with the added PA.
(2)$$\log_{10}\mathrm{suppression}=\log_{10}\mathrm{control}-\log_{10}\mathrm{sample}$$
where log_10_ control is the number of bacterial cells in the suspension itself and log_10_ sample is the number of bacterial cells in the suspension with the added PA.

#### 2.4.3. Long-Term Stability

The above-described analyses were performed within one month after he PA´s production. To verify the antimicrobial substances’ long-term stability in functionalized PAs, both antimicrobial tests were repeated nine months after the preparation of the PAs. Between the measurements, PAs were stored at room temperature (approx. 21 °C) separately from each other in a sterile plastic box.

### 2.5. Permeability Assay

The assay was based on filtering the *S. aureus* CCM 4516 suspension through a PA membrane placed in a specially designed glass filter apparatus (Section 3.3.1). The nanofiber membranes, sterilized with ethylene oxide at room temperature for 12 h cycles (Anprolene, Andersen Products, Clacton-on-Sea, UK) and evaporated for at least a week [41], were used as filters with a diameter of 5 cm (corresponding to the apparatus diameter; the diameter of the resulting filter surface was 4 cm). A total of 3 mL of bacterial suspension with OD adjusted to 1 McF was filtered through them. After the filtration, the procedures detailed below were performed.

#### 2.5.1. Filter Cultivation

The membranes were placed on Baird-Parker medium (BP, Merck, Darmstadt, Germany) as follows: one half of the PA with the filtration side up, and the other half with the bottom side up. After the incubation (24 h, 37 °C), bacterial growth on both sides of membranes was visually evaluated (*S. aureus* forms black-colored colonies at BP agar due to the reduction of potassium tellurite), compared, and recorded.

#### 2.5.2. Quantification of Bacterial Cells 

The bacteria both in the filtered suspensions and in the obtained filtrates were quantified as described in Section 2.4.2. At least three independent replicates were performed for each PA material. Then, log_10_ removal (CFU/mL) (Equation (3) [42]) and retention rate (Equation (4)) were calculated using the modified formula for inhibitory rate calculation [40].
(3)$$\log_{10}\mathrm{removal}=\log_{10}\mathrm{control}-\log_{10}\mathrm{sample}$$
where log_10_ control is the number of bacterial cells in the suspension itself and log_10_ sample is the number of bacterial cells in the suspension with the added PA.
(4)$$\mathrm{Retention}\;\mathrm{rate}\;\left[\%\right]=\frac{\mathrm{CFU}\;\left(\mathrm{control}\right)-\mathrm{CFU}\;\left(\mathrm{sample}\right)}{\mathrm{CFU}\;\left(\mathrm{control}\right)}\times 100$$
where CFU (control) is the CFU/mL in the filtered suspension and CFU (sample) is the CFU/mL in the final filtrate. When the final filtrate contained no cells, the sample was calculated to be <1.0 × 10^1^ CFU/mL.

#### 2.5.3. SEM Analysis of PA Filters

After the filtration of the bacterial suspension, PAs were rinsed with phosphate-buffered saline (PBS) and fixed with frozen absolute ethanol (Penta, Prague, Czech Republic) for 15 min at 4 °C. Then, the samples were dewatered with ethanol at increasing concentrations (60.0–99.8%). The PAs were dried for 24 h at room temperature and were observed by SEM under the conditions indicated in Section 2.2.

### 2.6. Statistical Analysis

All of the results are expressed as means and standard deviations for experiments performed in at least triplicates. The normality of the measured data was established by the Shapiro–Wilk test. The data were considered normally distributed at *p* > 0.05. Multiple comparisons of the data were determined by one-way analysis of variance (ANOVA), where the difference was assumed to be significant at the levels *p* ≤ 0.05 and *p* ≤ 0.01.

## 3. Results and Discussion

### 3.1. Nanomaterials

In total, 13 nanofibrous PA materials with different morphologies (fiber diameter, surface density, and air permeability) and concentrations of AgNO_3_ and CHX were tested (Table 1). PAs n. 1–n. 11 were characterized in the study of Lencova et al. [37], PAs n. 12 and n. 13 were characterized in this study. Examples of CHX PA structures are displayed in Figure 1. Among the most important parameters for evaluating a PA´s potential for practical use are the material morphology (incl. fiber diameter, surface density, porosity, air permeability, etc.) and homogeneity, indicating the overall material´s quality, mechanical properties, and water absorption. PA enables the production of homogenous structures with precise fiber diameters and similar pore sizes. These properties were confirmed for PAs n. 1–n. 11 [37] as well as the CHX PAs (Figure 1), where homogeneity, uniform, and evenly distributed fibers without any significant deviations or defects were observed. All of the properties are summarized in Table 1.

The permeability of the nanofibrous material is related to its fiber diameter, surface density, and pore size [43]. Permeability was evaluated using an air permeability test (Table 1). As the fiber diameter decreased and the surface density increased, this value decreased. The highest values (38.8 ± 2.1 L/m^2^/s) were reached for PA n. 4; in contrast, the smallest air permeability was measured for the material PA n. 3.

Suitable pore size, which provides optimal air circulation and guarantees the retention of unwanted microorganisms from the external environment, is essential for wound dressing materials. Pore size can vary enormously depending on the type of material used and the ability to prepare homogenous layers; it always depends on fiber diameter [44]. The pore size of PAs used in our study ranged from 200 to 500 nm, which is less than the size of most microbial cells. Therefore, we expected the membranes to retain most of the filtered bacteria. *S. aureus* used in this study forms round-shaped cells with an approx. size of 0.5–1.5 µm, which typically clump into grape-like clusters.

Besides the nanomaterial pore size, the other mechanical properties are important for their final use in practice; these are mechanical durability, strength, and maintaining their properties in various environments (dry or wet conditions). The analyzed PAs varied, and nanomaterials with higher surface density had the better mechanical properties and were more easily manipulated. The least suitable was PA 8% 2 g/m^2^, which ruptured easily and, therefore, should be used with other supporting materials. On the other hand, PA 15% 26 g/m^2^ had the best mechanical durability and could be applied alone.

The other property which cannot be neglected is PA´s water absorption, which is approximately 10% and is determined by the concentration of amide groups. The higher the ratio of CONH and CH_2_ groups, the greater the water absorption. The presence of the CONH bond causes PAs to absorb a certain amount of water, depending on their composition. With the increasing number of CH_2_ groups in the polymer, the amount of absorbed water decreases. Under moist conditions, PAs change their mechanical properties, and their impact strength and ductility increase, but their tensile and bending strength decrease [4,45]. During our measurements, it was confirmed that the thinner material absorbed the water faster. Therefore, all of the tested PAs differed in both their wettability and friability depending on their surface density. Based on these findings, we expected the nanomaterials to retain approximately 10% of the suspension during the experiments due to their absorption properties, and they did.

### 3.2. Inhibition Assays and Stability of Functionalized PAs

Nanomaterials´ antimicrobial properties are caused either by the material itself or by the addition of antimicrobial compounds. Whereas PA itself should have no antimicrobial effect [46], we expected AgNO_3_ or CHX functionalization to suppress bacterial growth, and this effect will be evident even after a long storage period. CHX has a bactericidal and bacteriostatic effect, especially against Gram-positive bacteria [30], and silver is toxic for both Gram-positive and Gram-negative bacteria [25,26]. The antimicrobial effect of the prepared PAs was measured using two methods:Inhibition zone;Inhibitory rate.

The inhibition zone method is a standard method used for the determination of antibacterial properties [47,48]; the inhibitory rate is not as common [49], but provides more accurate and evaluable results. The nonfunctionalized PAs (n. 1–n. 8) exhibited no inhibitory effect against *S. aureus* CCM 4516 in both inhibitory tests, and there was no statistical difference between them (*p* ≥ 0.05). Thus, PA n. 8 (PA 12% 13 g/m^2^) was chosen as a representative of nonfunctionalized PAs for further comparison.

#### 3.2.1. Inhibition Zone Method

Even though the bacterial growth was visibly suppressed under all functionalized PAs (Figure 2), the zones were only apparent for CHX PAs (Table 2). In the literature, CHX antimicrobial activity was primarily studied in connection with other material types. For example, da Silva et al. [50] tested a resin-modified glass-ionomer cement containing 1.25 and 2.50% CHX and proved its effectiveness against various microorganisms (*Streptococcus mutans*, *Lactobacillus acidophilus*, *Actinomyces israelii*, and *Candida albicans*). De Carvalho et al. [47] tested the antibacterial activity of cellulose acetate (CA) and polyethylene (PEO) nanofibers containing CHX; the nanofibers successfully inhibited *S. mutans* and *Enterococcus faecalis* growth. With PA nanofibers, Rysanek et al. [38] studied and confirmed the antibacterial activity of CHX PAs (0.1 wt% CHX) against various bacteria, including *S. aureus*.

AgNO_3_-functionalized nanomaterials are common. For example, in the study of Lala et al. [49], inhibition zones of AgNO_3_-functionalized CA, polyacrylonitrile (PAN), and polyvinylchloride (PVC) nanofibers were observed for *E. coli* and *P. aeruginosa*. Similar results were reported in studies focused on the antimicrobial effect of AgNO_3_-functionalized polyurethane nanofibers against *E. coli* and *Salmonella typhimurium* [51] and gelatin nanofibers against *S. aureus* and *P. aeruginosa* [48]. In the above studies, higher AgNO_3_ concentrations (5 wt% [49], 2–10 wt% [51], and 1–4 wt% [48]) were used than in our nanomaterials (0.1–0.5 wt%); therefore, it is not surprising that even when AgNO_3_ visibly suppressed the growth under the nanomaterial, the zones were not formed.

#### 3.2.2. Inhibitory Rate Method

PAs’ inhibition effect was further confirmed by the inhibitory rate method (Table 2). The average inhibitory rates ranged from 99.2 to 100.0% (*p ≤* 0.01) (Table 2). Except for the inhibitory rates, log_10_ suppression was determined; it ranged from 2.2 up to 8.9 CFU/mL, which means complete inhibition of bacterial growth. Inhibitory rates and log_10_ suppressions depended on the concentration of the active substances. With an increasing amount of active substances, the inhibitory rate/log_10_ suppression increased, and within the tested concentrations, the CHX PAs had a higher inhibitory effect. These results correspond with the outcomes of the inhibition zone method.

#### 3.2.3. Long-Term Stability

For an even more in-depth analysis, the long-term stability of the PAs was examined by both inhibition assay approaches nine months after their preparation. The obtained results (Table 2) were similar to the results measured within a month after PA production. No significant difference between them was found either by the inhibition zone method or inhibitory rate method (*p* ≥ 0.05). The prepared PAs are stable for at least nine months and do not change their antibacterial properties. Furthermore, this also confirmed that electrospinning produces stable and homogeneous nanomaterials. Even though these tests are crucial for confirming the materials´ long-term usability without losing their functionality, only a few studies determined their long-term effect. For example, Lala et al. [49] demonstrated the stability of CA-, PAN- and, PVC-functionalized nanofibers by the inhibition zone method after six months.

### 3.3. Permeability Assay

#### 3.3.1. Filtration Apparatus

For the testing, a special apparatus (Figure 3) that allowed the filtration of *S. aureus* CCM 4516 through the PAs was assembled. Its development consisted of various optimization steps, such as selecting a suitable material, designing a suitable shape, comparing different types of gaskets, and the possibility of vacuum generation to enable filtration through low-permeability materials. A similar apparatus was used in the study of Lubasova et al. [9] for *E. coli* filtration through PEO and PEO/purified soy flour (PEO/PSF) nanofibers.

Glass was chosen as the most suitable material, which can be reused due to easy sterilization with heat, chemicals (e.g., ethanol), or UV radiation. The shape of the apparatus enabled the suspension to be filtered through the nanomaterial either by gravity or using a vacuum (for example, by a pipetting balloon connected to a side outlet). Of the different tested variants of gaskets, one commonly available in stores proved to be the most suitable in conjunction with sealing rubber. In summary, the resulting apparatus is made up of two straight glass tubes with a round-shaped nanomaterial with a 5 cm diameter inserted between them, and the two tubes are then joined with a gasket (Figure 3).

#### 3.3.2. Retention of PAs for Staphylococci Cells

Due to the properties of nanomaterials, there is the general assumption that they will retain a certain amount of microorganisms depending on their morphology. A reliable filter function was expected for all of the prepared PAs. However, morphology parameters (fiber diameter and surface density) and functionalization can affect their final retention. Therefore, the following hypotheses were drawn up: H_1_**:** all of the PAs will retain most of the filtered bacteria; H_2_: fiber diameter and surface density will influence the retention; H_3_: the functionalization of PAs will support or increase their ability to retain bacteria.

The obtained permeability results for the PAs are summarized in Table 3 and Figure 4 and Figure 5. The enumerated log_10_ removals (CFU/mL) and retention rates ranged from 3.3 to 6.8 and from 96.7 to 100.0%, respectively. The majority of PAs achieved 2-log higher retention efficiency (except for PAs n. 1 and n. 4) than microfiltration membranes, which usually achieve 4 log_10_ removal during water filtration [39]. The cultivation of both sides of these PAs used as filters (Figure 4) confirmed that the bacterial cells were reliably captured, and no cells grew on the bottom side (except for PAs n. 1 and n. 4, where several cells grew on the bottom side). Furthermore, SEM analysis (Figure 5) of both sides of PA filters confirmed that bacterial cells did not pass through the materials. Bacterial cells were captured in the structure of the PAs; the images show both individual cells and clusters typical for *S. aureus* on the filtration side of nanomaterials. On the bottom sides of PAs, none or a few (in the case of PAs n. 1 and n. 4) retained cells were visible. However, retention rates > 96.7% were achieved for all materials (100% is achieved due to a reduction by two or three orders of magnitude [39]).

Our results are in agreement with other studies dealing with filtration through electrospun PA nanofibers. Most research was focused on the filtration of various inanimate materials, especially dust or aerosol particles. PA6 nonwovens varying in fiber diameter (50–150 nm) were studied in terms of their air filtration efficiency in the study of Zhang et al. [52]. The results showed PAs to have great potential for high-efficiency particulate air (HEPA) and ultra-low penetration air (ULPA) filters to capture particles 100 nm or less in diameter [52]. The high filtration efficiency of PAs for dust particles and aerosol particles (100–300 nm in diameter) was demonstrated in the studies of Guibo et al. [33] and Matulevicius et al. [34]. In terms of microbial retention, nylon-6 nanofibrous membranes (NFM) were proven to be effective filtration barriers for *S. cerevisiae*, *F. johnsoniae*, and *I. fluviatilis* in samples of water/broth and beer suspensions [10]. According to tests performed in a dead-end filtration assay, NFMs removed *S. cerevisiae* and bacterial mixtures completely. In the study of De Vrieze et al. [39], CHX PA nanofibers were tested and found to be effective membranes for water filtration with an efficiency of 4.0–4.2 log_10_ removal of microorganisms. Our results confirm the high bacterial retention of all the tested PAs (log_10_ removals from 3.3 to 6.8), especially that of PAs with a surface density of 5 g/m^2^ or higher (log_10_ removals from 6.4 to 6.8). Although the tests were only performed with one bacterial strain, *S. aureus* CCM 4516, we assume that the materials will be equally effective barriers for other staphylococci strains and other microorganisms of similar or larger size. Thus, PA nanofibers have great potential in all applications requiring high microbial retention.

#### 3.3.3. Influence of PA Morphology and Functionalization on Their Retention

Besides overall retention ability, we also focused on the impact of the PA´s morphology parameters (fiber diameter and surface density) and functionalization on their retention. Generally, fiber diameter is considered to be a crucial factor influencing the filtration efficiency of materials [44,53,54] and also plays a role in microbial interactions, mainly cell adhesion and biofilm formation [37,55]. Although the analyzed PAs were made up of three different percentages of PA solutions and thus differed in their fiber diameter (Table 1), there was no significant difference between their retention (*p* ≥ 0.05). This is in contrast to other studies where the smaller particles were filtered. Matulevicius et al. [34] revealed that PA electrospun from 8% solution with small fiber diameters (62–66 nm) had the highest filtration efficiency of the six tested polymer concentrations (8–26%). Guibo et al. [33] evaluated PAs electrospun from 13% solution as the most efficient of the tested nanomaterials, and even suggested them as references for the industry. In our experiments, staphylococci cells are larger than the size of the particles (300 nm) filtered in the above-mentioned studies, and, therefore, we suppose that fiber diameter plays a role when smaller particles are filtered.

Besides fiber diameter, surface density is also considered to be important for filtration performance [34,44]. The general assumption is that filtration efficiency increases with increasing surface density, and it decreases with increasing velocity of airflow [53]. Log_10_ removals (Table 3) differed within a set of materials prepared from the same polymer solution. Low surface density materials (e.g., PAs n. 1 and n. 4) might not have the same effect as thicker materials (e.g., PAs n. 3 and n. 7), probably due to their mechanical properties. Statistical analysis showed that surface density significantly influenced the retention of the PAs (*p* ≤ 0.01), and can be considered to be the most influential factor on a PA´s retention ability. The influence of surface density was mentioned by Lubasova et al. [9]; bacterial filtration efficiency by nanofibrous air filters from PEO increased (from 89.0 to 100.0%) with increasing surface density (from 1.0 to 5.0 g/m^2^). Matulevicius et al. [34] assumed that high filtration efficiency can be achieved by a suitable combination of fiber diameter and surface density.

In recent research, not only PA, but also PEO [56], polyacrylonitrile (PAN) [57], and polylactide/polyhydroxybutyrate (PLA/PHB) [58] electrospun filters were proven to be suitable filtration membranes. The common shortcoming of these studies is that after filtration, the captured bacteria contaminate the nanomaterial and might cause further biofilm formation [55,59,60]. This problem was observed and discussed in detail in the study of Rysanek et al. [38]. However, the biofilm formation can be overcome by proper nanomaterial functionalization, preventing bacterial cell growth [37,55,61]. PAs functionalized with AgNO_3_ and CHX were also tested. These materials achieved 100.0% bacterial retention (on average 6.8 log_10_ removal in CFU/mL, Table 3), and these outcomes did not significantly differ from nonfunctionalized PAs (*p* ≥ 0.05). Nevertheless, the bacterial cells captured by functionalized PAs were probably so damaged by AgNO_3_ or CHX that they did not grow on the PAs placed on the BP agar medium (Figure 4). This finding is supported by SEM (Figure 6), showing the damaged staphylococci cells on PA n. 10. Although the functionalization did not change the bacterial retention, it suppressed bacterial growth and prevented the subsequent production of a biofilm.

Our results support and expand the knowledge about the interactions between PAs and bacterial cells. To summarize, the following statements can be made:Functionalized PAs have antibacterial properties at all of the tested concentrations of AgNO_3_ and CHX;PA 12% 10 g/m^2^ CHX 4.0 wt% inhibits *S. aureus* CCM 4516 growth completely;Functionalized PAs are stable for at least nine months after their production;PA nanomaterials are effective bacterial barriers, which can be used in many applications;Surface density is the crucial morphological parameter influencing PAs’ ability to retain staphylococci cells;The functionalization of PAs with AgNO_3_ or CHX does not change filtration efficiency but makes PA usage safer due to the inhibition of bacterial growth.

In summary, the prepared PA nanomaterials exhibited significant microbiological benefits and can be used in medicine and filtration applications. They can serve as promising wound dressings or materials for the production of surgical masks with potential for the high retention of microorganisms and probably other undesirable particles. For further research, it would be appropriate to test PA nanomaterials when filtering other microorganisms, including viruses.

## 4. Conclusions

For the analysis of interactions between various PA nanofibrous materials and the bacterium *S. aureus* CCM 4516, inhibition and permeability assays were designed and verified. The antibacterial effect of PAs functionalized with AgNO_3_ and CHX was confirmed (an inhibition rate of up to 100.0% and 8.9 log_10_ suppression, which means complete inhibition of bacterial growth); this indicates that the effect of the antibacterial substances is not reduced during the electrospinning process. With increasing concentration of the active substance, the antibacterial efficiency of PAs increased. The most effective PA functionalized with 4.0 wt% of CHX exhibited the widest inhibition zones on the solidified medium and suppressed bacterial growth completely in the liquid medium. Furthermore, it was proven that functionalized PAs are stable and do not change their antibacterial properties for at least nine months after their production, without the need for specific storage conditions. Next, a glass filtration apparatus enabling the quick and easy filtration of a bacterial suspension through a nanomaterial was constructed. It was found that all of the PA nanofibers have a high bacteria retention ability reaching up to 100.0% and 6.8 log_10_ removal, i.e., all filtered cells. Fiber diameter and functionalization did not play a significant role in their efficiency, but surface density was found to be the most influential factor on PAs’ retention ability for staphylococci cells. These findings on the overall impact of PAs on *S. aureus* open the way for many medical applications. The properties of PA nanofibers enable their usage in medical devices, such as wound dressings, preventing the spread of infection, and providing highly effective bacterial barriers in other fields. Currently, PAs could be used especially in the development of effective surgical masks limiting the transmission of infectious agents.

## Figures and Tables

**Figure 1 nanomaterials-11-00480-f001:**
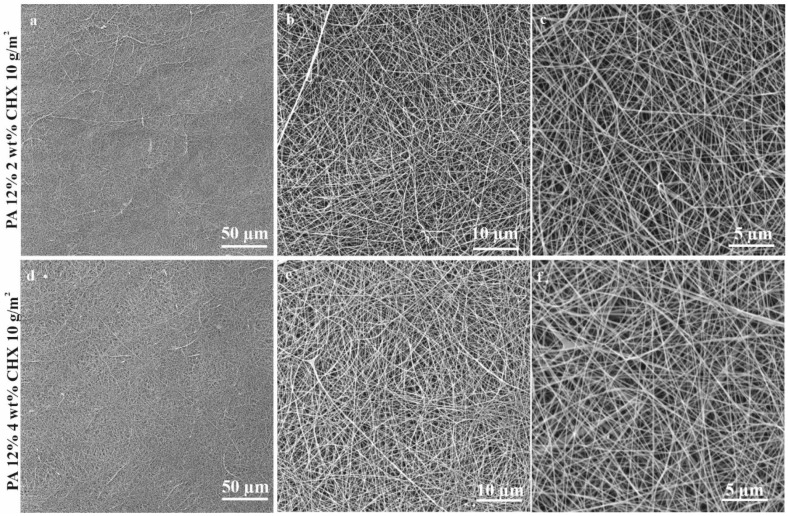
SEM images of used PA nanofibrous materials functionalized with CHX: 2 wt% CHX (**a–c**) and 4 wt% CHX (**d**–**f**) taken at magnifications of 1000× (**a**,**d**), 5000× (**b**,**e**), and 10,000× (**c**,**f**).

**Figure 2 nanomaterials-11-00480-f002:**
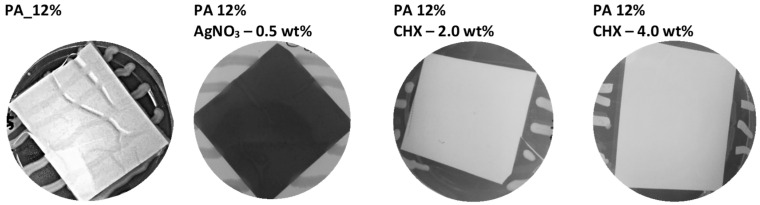
Examples of antibacterial activity test results for PAs n. 8 (PA_12%), n. 11 (PA 12% 0.5 wt% of AgNO_3_), n. 12 (PA 12% 2.0 wt% CHX), n. 13 (PA 12% 4.0 wt% CHX), and *S. aureus* CCM 4516. Around the CHX PAs, the inhibition zones are evident, whereas the other PAs show no inhibition zone. Only for the neat PA can bacterial growth on the material´s surface be observed.

**Figure 3 nanomaterials-11-00480-f003:**
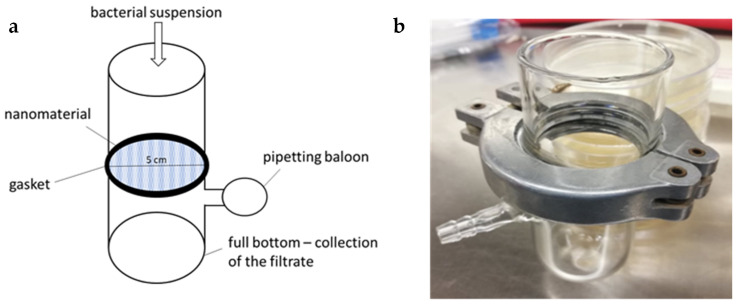
Filtration apparatus used for permeability assay; schema of the apparatus (**a**); photograph of the final device (**b**).

**Figure 4 nanomaterials-11-00480-f004:**
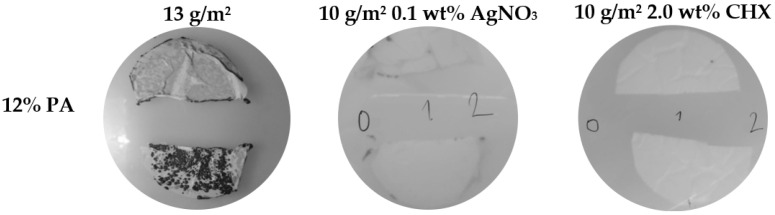
Examples of PAs cultivated at BP agar after the filtration of *S. aureus* CCM 4516; half of the PA at the top corresponds to the bottom side through which the suspension was not filtered, and half of the PA at the bottom corresponds to filtration side. The numbers (0, 1, 2, etc.) denote individual decimal dilutions of the final filtrates.

**Figure 5 nanomaterials-11-00480-f005:**
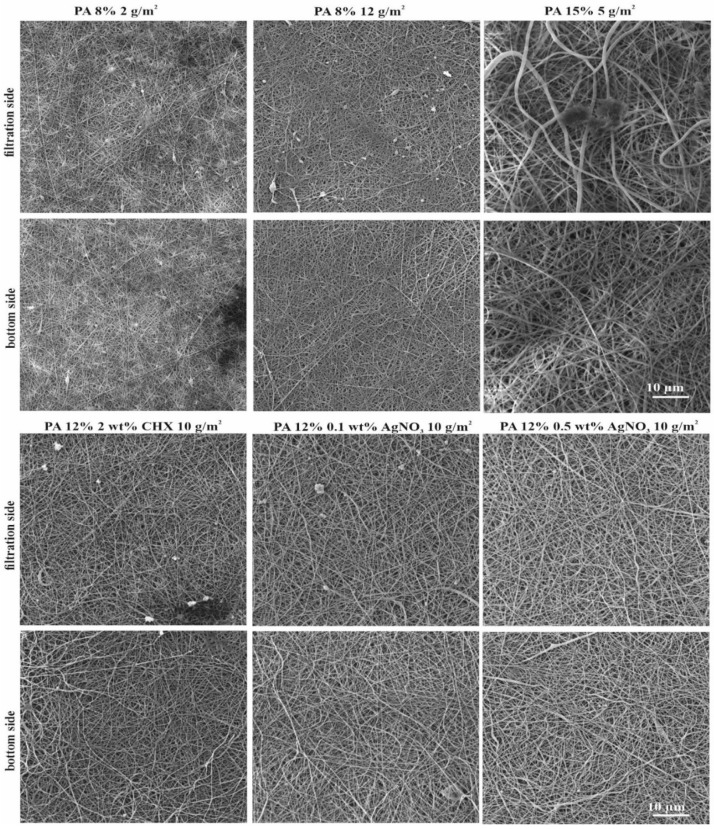
SEM figures of PAs after the filtration of *S. aureus* CCM 4516 (magnification of 5000×).

**Figure 6 nanomaterials-11-00480-f006:**
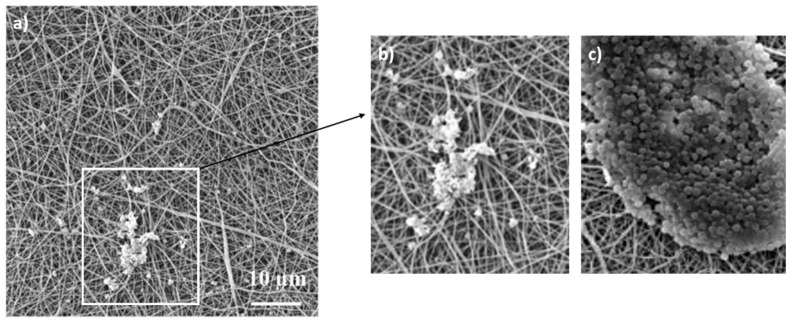
PA 12% 0.3 wt% AgNO_3_ 10 g/m^2^ after the filtration of *S. aureus* CCM 4516 (**a**,**b**); silver is toxic for bacterial cells and damages their shape (magnification of 5000×); (**a**) overall view, (**b**) detail of damaged bacterial cells, (**c**) clusters of undamaged *S. aureus* CCM 4516 cells on nonfunctionalized PA for a comparison with the silver one.

**Table 1 nanomaterials-11-00480-t001:** Characteristics of tested nanomaterials; * materials were characterized within our previous study [37] from which the parameters (antimicrobial compound, surface density, fiber diameter, and air permeability) were taken.

Material Number	Material	Antimicrobial Compound	Surface Density (g/m^2^)	Fiber Diameter (nm)	Air Permeability (L/m^2^/s)	Mean Pore Diameter (nm)
1 ^*^	PA 8% 2 g/m^2^	-	2.4 ± 0.3	87.7 ± 18.8	11.1 ± 1.2	222.7 ± 28.5
2 ^*^	PA 8% 5 g/m^2^	-	5.2 ± 0.5	87.7 ± 18.8	4.3 ± 0.3	222.7 ± 28.5
3 ^*^	PA 8% 12 g/m^2^	-	12.1 ± 0.1	87.7 ± 18.8	1.7 ± 0.2	222.7 ± 28.5
4 ^*^	PA 15% 2 g/m^2^	-	2.1 ± 0.1	236.2 ± 66.0	38.8 ± 2.1	477.5 ± 132.8
5 ^*^	PA 15% 5 g/m^2^	-	5.3 ± 0.4	236.2 ± 66.0	16.0 ± 11.7	477.5 ± 132.8
6 ^*^	PA 15% 11 g/m^2^	-	11.3 ± 0.7	236.2 ± 66.0	9.9 ± 3.1	477.5 ± 132.8
7 ^*^	PA 15% 26 g/m^2^	-	26.8 ± 0.8	236.2 ± 66.0	5.8 ± 2.1	477.5 ± 132.8
8 ^*^	PA 12% 13 g/m^2^	-	12.7 ± 0.3	151.7 ± 41.5	3.8 ± 0.1	395.8 ± 128.1
9 ^*^	PA 12% 10 g/m^2^	AgNO_3_ 0.1 wt%	10.6 ± 0.3	141.0 ± 48.1	3.5 ± 0.2	331.5 ± 149.2
10 ^*^	PA 12% 10 g/m^2^	AgNO_3_ 0.3 wt%	9.3 ± 0.3	139.5 ± 45.8	4.3 ± 0.2	330.0 ± 145.0
11 ^*^	PA 12% 10 g/m^2^	AgNO_3_ 0.5 wt%	9.3 ± 0.2	108.0 ± 23.7	4.6 ± 0.2	379.0 ± 167.6
12	PA 12% 10 g/m^2^	CHX 2.0 wt%	10.0 ± 0.7	112.8 ± 20.2	3.3 ± 0.3	361.7 ± 177.9
13	PA 12% 10 g/m^2^	CHX 4.0 wt%	9.7 ± 0.6	108.4 ± 21.9	3.4 ± 0.2	401.8 ± 171.4

**Table 2 nanomaterials-11-00480-t002:** Average inhibition zones, inhibitory rates, and log_10_ suppression of PAs n. 8–n. 13 for *S. aureus* CCM 4516 both shortly (1st analysis) and after nine months (2nd analysis) after PA preparation. Inhibition zones were measured from the edge of the material. “N” means that the bacterium grew under the PA; “0” means that the bacterium did not grow under the PA, but there were no inhibition zones around. *p*-values confirm no significant differences between the 1st and 2nd measurement.

Material	Inhibition Zones (mm)		Inhibitory Rate (%)		log_10_ Suppression (CFU/mL)	
	1st Analysis	2nd Analysis	1st Analysis	2nd Analysis	1st Analysis	2nd Analysis
PA 12% 13 g/m^2^	N	N	2.0 ± 1.0	3.0 ± 1.4	0.0 ± 0.0	0.0 ± 0.0
PA 12% 10 g/m^2^ AgNO_3_ 0.1 wt%	0.0 ± 0.0	0.0 ± 0.0	99.2 ± 0.1	99.5 ± 0.3	2.2 ± 0.2	2.5 ± 0.4
PA 12% 10 g/m^2^ AgNO_3_ 0.3 wt%	0.0 ± 0.0	0.0 ± 0.0	100.0 ± 0.0	100.0 ± 0.0	8.0 ± 0.1	7.5 ± 0.6
PA 12% 10 g/m^2^ AgNO_3_ 0.5 wt%	0.0 ± 0.0	0.0 ± 0.0	100.0 ± 0.0	100.0 ± 0.0	8.7 ± 0.3	8.6 ± 0.4
PA 12% 10 g/m^2^ CHX 2.0 wt%	1.8 ± 0.1	1.6 ± 0.2	100.0 ± 0.0	100.0 ± 0.0	8.9 ± 0.0	8.9 ± 0.0
PA 12% 10 g/m^2^ CHX 4.0 wt%	2.9 ± 0.2	2.4 ± 0.3	100.0 ± 0.0	100.0 ± 0.0	8.9 ± 0.0	8.9 ± 0.0
*p*-value (α = 0.05)	0.87		0.98		0.99	

**Table 3 nanomaterials-11-00480-t003:** Average log_10_ removals, retention rates, and visual evaluation of growth on Baird-Parker (BP) agar (“+” means bacterial growth was detected, “−“ means bacterial growth was not detected) of PAs for *S. aureus* CCM 4516.

Material Number	Material	log_10_ Removal (CFU/mL)	Retention Rate (%)	Growth on BP Agar	
				Filtration Side of PA	Bottom Side of PA
1	PA 8% 2 g/m^2^	3.3 ± 1.6	96.7 ± 4.6	+	+
2	PA 8% 5 g/m^2^	6.4 ± 0.5	100.0 ± 0.0	+	−
3	PA 8% 12 g/m^2^	6.7 ± 0.0	100.0 ± 0.0	+	−
4	PA 15% 2 g/m^2^	4.3 ± 0.3	100.0 ± 0.0	+	+
5	PA 15% 5 g/m^2^	6.7 ± 0.0	100.0 ± 0.0	+	−
6	PA 15% 11 g/m^2^	6.7 ± 0.0	100.0 ± 0.0	+	−
7	PA 15% 26 g/m^2^	6.5 ± 0.2	100.0 ± 0.0	+	−
8	PA 12% 13 g/m^2^	6.4 ± 0.2	100.0 ± 0.0	+	−
9	PA 12% 10 g/m^2^ AgNO_3_ 0.1 wt%	6.8 ± 0.1	100.0 ± 0.0	−	−
10	PA 12% 10 g/m^2^ AgNO_3_ 0.3 wt%	6.8 ± 0.1	100.0 ± 0.0	−	−
11	PA 12% 10 g/m^2^ AgNO_3_ 0.5 wt%	6.8 ± 0.1	100.0 ± 0.0	−	−
12	PA 12% 10 g/m^2^ CHX 2.0 wt%	6.7 ± 0.1	100.0 ± 0.0	−	−
13	PA 12% 10 g/m^2^ CHX 4.0 wt%	6.8 ± 0.1	100.0 ± 0.0	−	−

## Data Availability

Data are available on request to the corresponding author.

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
