# Peer review of "Benefits of Polyamide Nanofibrous Materials: Antibacterial Activity and Retention Ability for Staphylococcus Aureus"

_nanomaterials, 2021, doi:10.3390/nano11020480_

Round 1

Reviewer 1 Report

Line 147-148 - "The PCA plates were first inoculated by spreading five horizontal stripes of a bacterial suspension at a sufficient distance from each other" -  please detail what it means "sufficient distance from each other"? Did you use a standard method (adapted or not to the experiment)? For a correct interpretation of the inhibition zone it is necessary that the bacterial growth on the medium surface to be uniform.

Conclusions - I suggest a better emphasis on the medical applications of the  PA nanofibrous materials and the advantages of their use.

Author Response

In lines 147 – 149, we detailed what “sufficient distance from each other” means. For the inhibition zones method, we used a standard method adapted to the experiments. The PCA plates were first inoculated by spreading five horizontal stripes of a bacterial suspension at a sufficient distance (approx. 1 cm) from each other, and a piece of a sterile PA (5x5 cm) was then placed onto the inoculated surface. The spreading of horizontal stripes of a bacterial suspension was chosen as a more illustrative procedure (than spreading a suspension all over the agar surface) for demonstrating the antibacterial effect of the PAs. Also, the inhibitory rate method, which is more accurate, was used to confirm the antibacterial effect of functionalized PAs.

In conclusions (lines 510 – 514), we emphasized more the medical applications of the PAs. The findings on the overall impact of PAs on S. aureus open the way for many medical applications. The properties of PA nanofibers enable their usage in medical devices, such as wound dressings, preventing the spread of infection and providing highly effective bacterial barriers in other fields. Currently, the PAs could be used especially in the development of effective surgical mask limiting the transmission of infectious agents. 

Reviewer 2 Report

This paper reports the bacterial filtration capability of polyamide nanofibres; this is a highly important piece of work which is of immense relevance to current medical emergencies, and morever is conducted with a level of thoroughness and care not normally seen.  It is therefore a showcase paper from a procedural point of view.  For these two reasons, immediate publication is recommended.

Author Response

Thank you very much for your positive comment.

Reviewer 3 Report

This work focuses on the fundamental interactions between nanomaterials and microorganisms. To test antimicrobial properties and retention ability, 13 electrospun polyamide (PA) nanomaterials with different morphology and functionalization with various concentrations of AgNO3 and chlorhexidine (CHX) were analyzed. Staphylococcus aureus CCM 4516 was used to verify the designed nanomaterials´ inhibition and permeability assays.

In this work PA nanomaterials exhibited significant microbiological 

benefits and can be used in medicine and filtration applications. They can serve asp romising wound dressings or materials for the production of medical drapes with potential for the high retention of microorganisms and probably other undesirable particles.

I suggest the authors to stress al little more their work and its novelty. They can end the introduction of this manuscript focusing on their work and focus on their work.

Some syntax and English issues need to be resolved.

Could the authors comment on the reusability of their samples? Could we re-use them, or we are talking about single use filters?

I suggest this manuscript to be published after minor revisions. 

Author Response

We modified the introduction (lines 111-113) and stressed the novelty of our research. To our knowledge, we are the first to published basic research about the retention ability of PA nanomaterials for a clinically relevant bacterium and thus increased their usage in several medical applications. Furthermore, the presented comprehensive design of testing microbiological safety and stability of PA nanofibrous materials is unique.

Before submitting the manuscript, English proofreading was provided.

All tested PA nanofibers were used as single-use filters. However, the potential of their reusability is considerable, and these analyzes will be included in the follow-up study.
